# Regional Living Conditions and Individual Dietary Characteristics of the Russian Population

**DOI:** 10.3390/nu15020396

**Published:** 2023-01-12

**Authors:** Sergey A. Maksimov, Natalia S. Karamnova, Svetlana A. Shalnova, Galina A. Muromtseva, Anna V. Kapustina, Oksana M. Drapkina

**Affiliations:** National Medical Research Center for Preventive Medicine of the Ministry of Healthcare of the Russian Federation, 10 Petroverigsky Ln., 101990 Moscow, Russia

**Keywords:** diet, living environment, dietary patterns, characteristics of regions, Russia

## Abstract

The goal of our study was to examine the effects of the regional characteristics of the living environment on individual a priori and a posteriori dietary patterns of the Russian population. For the analysis, we used cross-sectional data from the Epidemiology of Cardiovascular Diseases in the Regions of the Russian Federation study from 2013–2014. The sample included 18,054 men and women 25–64 years of age from 12 regions. Based on the frequency of consumption of basic foods, four a posteriori empirical dietary patterns (EDPs), along with an a priori cardioprotective dietary pattern (CPDP), were identified. To describe the regional living environment, five regional indices were used. Adherence to the meat-based EDP was directly associated with deterioration of social living conditions and a more northerly location for the region of residence. The probability of a CPDP increased with greater deterioration of social living conditions, aggravation of demographic crises, and higher industrial development in the region, as well as with declines in the economic development of the region, income, and economic inequality among the population. We detected several gender-dependent differences in the associations established. The patterns revealed reflect the national dietary preferences of Russians, and the regional indices characterize the effect of the living environment.

## 1. Introduction

Global differences in the structure of human nutrition by region and country are apparent worldwide [1,2]. Furthermore, even within groups of countries with similar cultures, there are significant differences in the structure of human nutrition. Previously, the European Prospective Investigation into Cancer and Nutrition (EPIC) study demonstrated geographical differences in food consumption among European countries: fruit and vegetables [3], fish [4], dairy products [5], and meat [6]. These findings were also confirmed by later studies [7]. Going further down the geographic scale gradient, the EPIC study [8] and other nationwide studies [9,10,11] demonstrated gradations of variation in food consumption not only between entire countries but also between different regions within particular countries.

However, it is clear that specific geographic locations are not predictors of different diets per se. The systemic approach that considers the interaction between the genetic, biological, behavioral, psychological, social, and environmental factors of health assigns a significant role to the so-called social production of diseases [12,13]. The theoretical concept of environmental impact is based on the fact that the essential characteristics of life (social inequality, economic inequality) influence intermediate factors. The latter, in turn, affect psychoemotional and behavioral predictors, which may be direct (smoking) or indirect (physical activity or diet) health risk factors [13]. In the field of dietetics, environmental factors can have a specific effect, since following the behavior of others is adaptive and social norms inform people about the behavior that may be optimal in a particular environment: “If everyone else is doing it, I should probably do it too” [14,15].

At a small geographic scale (city blocks, districts, census tracts), environmental impact on individual diet, expressed in specific characteristics, is conceptually justified [16]. Numerous studies have demonstrated the impacts of various factors on dietary preferences; e.g., the population characteristics of areas, including the socioeconomic status [17,18,19] and unemployment rate [20]. However, only a few studies have considered the impact of environmental factors on nutrition at the scale of large national regions (state, province, etc.). A Swiss study confirmed substantial differences in dietary patterns between three language regions in the country; i.e., the predictors in this case were German, French, or Italian cultural living environments [21]. In a Canadian study of adolescents, the Gini index and the mean household income in the province were considered as environmental predictors of the quality of breakfasts (as a behavioral factor of child health), but no associations were revealed [22]. However, it is clear that understanding the environmental predictors of eating behavior provides an opportunity to recognize the underlying causes of the uneven distribution of population health and, accordingly, could offer promising approaches to diminishing such dissimilarities.

In Russia, regional aspects of nutrition were most fully studied within the framework of the epidemiological study Epidemiology of Cardiovascular Diseases in the Regions of the Russian Federation (ESSE-RF) conducted in 2013–2014. The study established a high degree of coincidence for the a posteriori regional dietary pattern and the nationwide Russian pattern in just 7 out of 13 participating regions [23]. In addition, regional differences in the a priori cardioprotective dietary pattern of healthy eating have been demonstrated [24]. These studies demonstrated that the traditional diet of the Russian population is fairly diverse. It includes both healthy and unhealthy foods. Among the Russian a posteriori dietary patterns, in terms of the assortment of food, the prudent model mainly corresponds to the healthy or balanced Western European models, and the combination of salt-rich and meat-based models matches a Western European high-salt dietary pattern. It is worth mentioning that adherence to the prudent model was associated with a decrease in the likelihood of risk factors for chronic non-communicable diseases (arterial hypertension, obesity, hyperglycemia, hypertriglyceridemia, hypercholesterolemia), while adherence to the salt-rich and meat-based models, in contrast, was accompanied by an increase for such risks [25].

However, in this study, as in most similar studies abroad, the regional characteristics of the living environment of the population, which could possibly affect individual food preferences, were not examined. Consequently, the goal of our study was to examine the effects of the regional characteristics of the living environment on individual a priori and a posteriori dietary patterns in the Russian population.

## 2. Materials and Methods

### 2.1. Study Sample

For the analysis, we used data from the cross-sectional epidemiological study Epidemiology of Cardiovascular Diseases in the Regions of the Russian Federation (ESSE-RF) conducted in 2013–2014 [26]. In total, 21,923 people aged 25–64 years old from 13 regions of the Russian Federation participated in the study. The Kish selection grid was used to form the sample, providing systematic, multistage, random sampling according to the territorial principle on the basis of medical institutions. The study was approved by the ethics committees of three federal centers: the State Research Center for Preventive Medicine, the Russian Cardiology Research and Production Complex, and the V.A. Almazov Federal Medical Research Center. The study was performed in compliance with the ethics of good clinical practice (GCP) and the principles of the Declaration of Helsinki. Written informed consent was obtained from all participants prior to their enrollment in the study. The response rate for the survey was approximately 80%, varying across the study regions.

Participants with missing data on diet (n = 2403) and smoking (n = 6) were excluded from the final sample. In addition, a subsample from St. Petersburg (n = 1460 subjects) was excluded as being significantly different in its regional characteristics from the other 12 regions. The city of St. Petersburg is classified in the Russian Federation as a separate administrative territorial unit, while the other 12 regions are large areas including both urban areas and countryside. Therefore, the final analytical sample comprised 18,054 people from 12 regions, including 6814 men and 11,240 women.

### 2.2. Methods for Assessing and Analyzing the Dietary Patterns

Dietary preferences were assessed based on the results of face-to-face interviews on the frequency of consumption of ten food groups: meat (beef, pork, lamb, etc.), fish and seafood, poultry (chicken, turkey, etc.), sausages and offal (tongue, liver, heart, etc.), pickled foods, cereals and pasta, fresh fruit and vegetables, legumes (beans, lentils, peas, etc.), sweets and pastries (candies, jams, cookies, etc.), and dairy products (milk, kefir, and yogurt; high-fat varieties: sour cream, cream, cottage cheese, and cheese). Since the goal of our study was to analyze the impact of regional living conditions on both a priori and a posteriori dietary patterns, the nutritional characteristics were determined using two corresponding methodological approaches [27].

For a posteriori assessment of nutrition, we used four empirical dietary patterns (EDPs) previously identified via the principal component method [23]: a prudent EDP involving high consumption frequency for cereals and pasta, fruit and vegetables, sweets and confectionery, and dairy products; a salt-rich EDP including frequent consumption of sausages and pickled products; a meat-based EDP presenting high consumption frequency for red meat, fish and seafood, and poultry; and a mixed EDP incorporating frequent consumption of fish and seafood, pickled foods, and legumes. For each study participant, an individual assessment of predisposition to each of the four EDPs was calculated with a normal distribution, a mean equaling 0, and a standard deviation equaling 1. Thus, the a posteriori assessment was characterized by an individual quantitative coefficient of predisposition to each of the four EDPs.

For the a priori assessment of nutrition, a cardioprotective dietary pattern (CPDP) was used that included the presence of the following four eating habits: daily consumption of vegetables and fruit, eating fish at least one to two times a week, exclusive use of vegetable oils for cooking, and consumption of dairy products with reduced fat content. A detailed description of the CPDP has been presented elsewhere [24]. Since the four-component CPDP had rather strict nutritional requirements and its prevalence in the general population was low (just 6.5% of the general population), the three-component CPDP was also considered for confirmatory analysis. The three-component CPDP was recorded for the presence of any three of the four eating habits listed above. Its prevalence in the general population was 32.6%, which allowed confirmation or refutation of the regional associations of the four-component CPDP with a high degree of confidence.

### 2.3. Individual Covariates

Characteristics that could potentially affect the examined relationships were used as covariates for individual variables. Gender, age, and place of residence (urban/rural) were identified from the survey data. Some data were obtained through interviews: education level (higher/other than higher), family status (has a family/does not have a family), professional employment (employed/unemployed), and smoking status (never smoked, quit smoking, current smoker). Self-assessment of the presence of various diseases that could potentially affect individual dietary patterns was also taken into account (diseases of the liver, gallbladder, and gastrointestinal tract; gastric and duodenal ulcers; type 1 and type 2 diabetes mellitus). Body mass index was calculated as the body mass (in kg) divided by the square of the body height (in m2).

### 2.4. Regional Variables

To describe regional living conditions, an integral index assessment was used, which was developed previously using principal component analysis [28]. To identify regional indices, publicly available data from the official website of the Federal State Statistics Service of Russia (www.gks.ru, accessed on 18 December 2022) for 2010–2014 were used. In total, we identified five regional indices, which were considered as quantitative indicators reflecting a negative (negative index values) or positive (positive index values) ratio in a particular region.

The socio-geographical index combined ten characteristics: (a) mean per capita consumption of vodka; (b) mean per capita consumption of wine; (c) mean per capita consumption of low-alcohol beverages; (d) mean per capita consumption of cognac; (e) mean annual air temperature (negative load on the factor); (f) forested area in the region; (g) per capita crime rate; (h) the geographical latitude of the regional center location; (i) the proportion of dilapidated housing; (j) proportions of school students studying in the morning and afternoon shifts. In general, an increase in this index value reflected the deterioration of the social environment and, simultaneously, a more northerly location for the region, implying worse climatic conditions.

The demographic index was formed of five characteristics: (a) natural population growth (negative load); (b) total fertility rate (negative load); (c) crude death rate; (d) the proportion of people at retirement age among the population; (e) mortality from respiratory diseases. Growth in this index value indicated a greater demographic crisis in the region due to depopulation and changes in the population age structure in favor of older age groups.

The industrial index included eight characteristics: (a) the amount of mining; (b) the volume of electricity generated; (c) mortality from tuberculosis; (d) the death rate from infectious diseases; (e) the mortality rate from external causes; (f) the proportion of people in the region working in hazardous conditions; (g) population numbers in the region; (h) atmospheric emissions. An increase in this index value was indicative of growth in the industrial development in the region, primarily due to mining and energy production, with corresponding exposure of the workforce and inactive population to adverse anthropogenic effects.

There were five components incorporated into the mixed index: (a) the number of workers in fish farms; (b) the mean per capita amount of paid services; (c) the mean per capita number of motor vehicles; (d) the ratio of men to women (negative load); (e) the geographical longitude of the regional center location. The mixed index was the most difficult to interpret since, in general, it resulted in unambiguous characterization based on the entirety of its constituent indicators.

The economic index was formed of five characteristics: (a) volume of per capita retail trade; (b) mean per capita household consumption; (c) the Gini index; (d) mean per capita income of the population; (e) the level of the manufacturing industries in the region. An increase in this index value reflected growth in economic development, income, and population economic inequality in the region.

### 2.5. Methods of Statistical Analysis

Categorical variables are presented as frequencies, while quantitative variables are presented as means and standard deviations. The examined data constituted a complex two-level sample with individual and regional characteristics. Hence, a generalized estimation equation with robust standard errors, taking into account the nested structure of the data (study subjects in regions, n = 12), was used to measure associations. Since the EDPs (prudent, salt-rich, meat-based, and mixed) represented normally distributed scores, their associations with regional indices were estimated via linear regression models. The CPDP represented a binary response; consequently, a logistic regression model was used for it. In the linear models, associations were expressed by the B-coefficient and the level of statistical significance (*p*-value); in the logistic models, they were characterized by the odds ratio (OR) and 95% confidence intervals (CIs). All statistical models were adjusted for individual covariates: gender, age, place of residence, education level, family status, professional employment, smoking status, body mass index, and the presence of diabetes mellitus, gastric or duodenal ulcers, and diseases of the liver, gallbladder, and gastrointestinal tract. Regional indices were entered into the regression models all together. During the pilot study, several interactions between gender and regional indices were found; therefore, in addition to the analysis of the general sample, we also performed separate analyses for men and women. To compare the significance of the regional indices with confirmed individual predictors of dietary patterns, the values of the effect parameters in Wald chi-squared models were calculated. The critical level of statistical significance was assumed to be the value of 0.05. All statistical procedures were carried out in SPSS version 22 (IBM Corporation, New York, NY, USA).

## 3. Results

The analyzed sample was dominated by urban residents who had a family, had education other than higher education, were professionally employed, and were non-smokers who had never smoked (Table 1). Diseases of the liver, gallbladder, and gastrointestinal tract were typical for 38.2% of them; gastric or duodenal ulcers were characteristic for 12.9%; diabetes mellitus was present in only 4.6% of the sample. The mean age in the sample was 46.4 ± 11.6 years, while the BMI was 28.1 ± 5.9 kg/m^2^. For many parameters, we established differences between male and female study participants. Among women, we observed lower proportions of participants who had a family, were professionally employed, and were smokers, whereas the proportions of those with diseases of the liver, gallbladder, and gastrointestinal tract were higher.

Table 2 shows the associations for the complete models in the general sample, including individual predictors and regional indices. In addition, Appendix A present the complete models separately for men and women.

Individual adherence to the prudent EDP increased with growth in the mixed index, and the B-coefficient was −0.054 at *p* < 0.001. In addition, in women but not in men, adherence to the prudent EDP increased with the deterioration of social living conditions for the population and a more northerly location for the region of residence (socio-geographical index), and the coefficient was 0.131 at *p* = 0.002 (Appendix A).

The inverse association between individual adherence to the salt-rich EDP and the mixed index was typical solely for men (the B-coefficient value was −0.039 at *p* < 0.001) and not for women (Appendix A).

Adherence to the meat-based EDP was directly associated with the deterioration of the social living conditions for the population and a more northerly location for the region of residence (socio-geographical index), and the coefficient was 0.103 at *p* = 0.046.

Associations with all regional indices were typical for the CPDP. The probability of a CPDP was directly associated with the socio-geographical (OR = 1.23; 95% CI: 1.08–1.41), demographic (OR = 1.40; 95% CI: 1.29–1.52), industrial (OR = 1.09; 95% CI: 1.04–1.15), and mixed (OR = 1.16; 95% CI: 1.12–1.19) indices. In other words, the probability of this dietary pattern increased with the deterioration of social living conditions, aggravation of demographic crises, and higher industrial development in the region. An inverse association with CPDP probability was observed for the economic index (OR = 0.90; 95% CI: 0.81–0.99); that is, with an increase in the economic development in the region, income, and economic inequality among the population, the probability of the CPDP declined. An additional analysis of the three-component CPDP demonstrated a lower number of statistically significant associations (Appendix A), although they had the same direction: a direct relationship with the socio-geographical, demographic, and mixed indices. Hence, overall, the less stringent three-component CPDP confirmed the associations of the strict four-component CPDP.

Table 3 presents the significance of individual and regional predictors of dietary adherence. The contributions of various individual predictors were strong, which was in agreement with numerous published sources on the dependence of dietary patterns on individual biological and socioeconomic characteristics and health status. Against the background of confirmed individual predictors, the significance of regional indices for individual adherence to EDPs (prudent, salt-rich, meat-based, and mixed) was rather low. We noted a high contribution from the regional mixed index solely for the prudent EDP and mixed EDP: 26.6 and 19.3, respectively. The contributions of the regional indices (especially the mixed and demographic indices) were rather high for the CPDP: 84.2 and 64.2, respectively.

## 4. Discussion

Our results highlighted the impacts of the regional characteristics of the living environment on individual adherence to various dietary patterns, with some gender-based distinctions. EDPs (prudent, salt-rich, mixed) were predominantly associated with the mixed index. The latter was, regrettably, the most difficult to interpret. In addition, adherence to the prudent EDP (for women) and meat-based EDP was associated with the socio-geographical index. Commitment to the CPDP was connected to all regional indices.

From the standpoint of assessing diet as a behavioral risk factor affecting health, the dependence of population dietary patterns on regional living conditions was somewhat surprising. As a rule, studies of the dependence of eating habits on the living environment at the scale of districts demonstrate the negative impact of the low socioeconomic status of districts (high unemployment rate, low income, low education level, and low-paid jobs). For instance, a 2015 systematic review [18] discovered a positive association between low consumption of fruit and vegetables and living in disadvantaged neighborhoods compared to prosperous areas in two studies on the topic [29,30]. These findings were also confirmed by more recent studies. For example, in the Finnish population, the cumulative socioeconomic disadvantage of neighborhoods was associated with poorer nutrition, whereas high population density exhibited a relationship to better adherence to dietary recommendations [31].

In our study, adherence to the healthiest diet (the CPDP) increased with the deterioration of social and economic living conditions, aggravation of the demographic situation, industrial development, and increased economic inequality. In women, similar associations with social living conditions were exhibited for the prudent EDP, which was the healthiest of all the EDPs. Moreover, adherence to the meat-based EDP involving frequent consumption of unprocessed red meat, fish and seafood, and poultry was higher in socially disadvantaged regions as well. The divergence from the results of other studies utilizing the framework of the health geography concept has two potential explanations. First, other studies operated with a smaller spatial scale for the representation of the living environment; accordingly, the comparability of the effects at different geographic scales was one of the probable sources for detecting the multidirectional impact [32,33,34]. It is likely that patterns that are obvious at the level of districts do not manifest themselves—or manifest themselves differently—at larger spatial scales (e.g., national regions).

Furthermore—and, in our opinion, this was the crucial issue—the other studies were carried out with samples of populations representing countries with a high level of economic development and, in general, one cultural group (Western), while our results were to be expected from the standpoint of national Russian dietary preferences. A systematic analysis of dietary surveys around the world [1] showed that Russia fell into the category of rather low fruit consumption (100–124 g/day, grade four on the ten-grade scale) in the world rankings, but this was offset by higher consumption of vegetables (200–250 g/day, grade seven). Consequently, the cumulative intake of fruit and vegetables in Russia was relatively high.

Furthermore, Russia is characterized by frequent consumption of dairy products and fruit juices [35]. In addition, Russia has the highest cumulative intake of animal products among the most populous countries (5.8 servings daily) [36]; e.g., there is traditionally high intake of unprocessed red meat (70–79 g/day, grade eight), processed meat (30–34 g/day, grade seven), and fish and seafood (30–34 g/day, grade six) in Russia [1]. Overall, in Russia, against the background of a high frequency of cholesterol consumption with food, there is a rather high level of fat intake from vegetable products and seafood [37].

Since Russia belongs to the group of countries with an average level of economic development, it is obvious that frequent consumption of such products is available to all segments of the population, including low-income households; that is to say, pursuing a healthy diet in Russia is not particularly expensive. For example, according to the PURE study, the absolute cost of various fruits in countries with an average level of economic development is the highest, while the cost of vegetables in such countries is the lowest (adjusted for parity of wholesale purchase prices) [38]. Another specific Russian feature is the traditional development of personal subsidiary farms, providing high all-season availability for vegetables, berries, and some varieties of fruits (primarily apples) for all segments of the population [39,40]. For instance, as of 2020, 27.4% of the total output of agricultural production (in actual prices) across the entire Russian Federation was produced in personal subsidiary farms [39]. In certain federal districts, this figure exceeded 40%, reaching 60–70% in some regions. Furthermore, personal subsidiary plots and, consequently, augmented consumption of vegetables, berries, and some fruit varieties are common among the most vulnerable population segments.

Thus, adherence to healthy eating stereotypes and high intake of red meat are traditional and rather affordable for the Russian population. The results of our study confirmed that such traditional stereotypes of eating behavior were especially characteristic of people in socially, economically, demographically, and environmentally unfavorable living conditions (regions). Such regions host much higher proportions of people from the low-income, poorly educated, elderly, and socially vulnerable categories of the population. In adverse living environments, these population categories seem to prefer traditional stereotypes of eating behavior. In regression models, such patterns should supposedly manifest themselves via the effects of individual socioeconomic—rather than regional—characteristics on adherence to dietary patterns. Consequently, we customized our analysis to individual characteristics (including those of a social nature) that could affect adherence to dietary patterns; even then, the effect of regional living conditions was statistically significant. This finding implied that, regardless of individual characteristics, the living environment affected individual eating behavior. Therefore, the behavioral habits of the predominant part of the populations in disadvantaged regions were transferred to the entire populations of such regions, while individual differences were leveled off to a certain extent.

The theoretical substantiation of such environmental conditioning of nutrition is provided by numerous studies on the impact of alleged dietary norms on human eating behavior. As with other behaviors, perceived eating norms can act as behavioral cues of utility: “Everyone else behaves this way for some reason, so I should probably behave like this too” [14,41,42]. Despite the fact that behavioral cues exhibit the most pronounced influence on family members and socially close individuals [43,44], the perceived norms at the population level also affect health behaviors, including dietary preferences [14,45]. An interesting example of such influence is presented by Guendelman et al. [43]: two experiments confirmed that, for immigrants of Asian origin, the desire to prove their American identity motivated them to eat more typical American foods, such as fast food.

It is worth noting that the augmented adherence to the meat-based EDP with growth in the values of the socio-geographical index could be associated with the geographic component of the index rather than its social aspect alone. If we focus our attention on the geographic component, then it can be seen that the adherence to the meat-based EDP increased with a more northerly location for the region. Other studies have also revealed a north–south gradient in the consumption of meat products [6], and animal products in general [46,47], which was associated with behavioral adaptation to cold environmental conditions [48,49].

It is not so easy to exactly identify the decisive regional factor (social matters or climatic/geographic conditions) because there is a strong interaction between the two. Hence, this issue will be a task for our further analyses.

One of the advantages of our study was that, for the first time, an attempt was made to consider the dependence of individual dietary patterns on a wide range of living conditions for the population at the scale of national regions. The study was carried out with a large representative sample of the Russian population that included 12 regions covering all climatic and geographical zones nationwide (Central Region, Southern Region, Volga Region, Urals, Siberia, Far East), with the exception of the Far North.

However, our results were somewhat limited. The main limitation was the use of the short version of the dietary pattern questionnaire, which lacks serving sizes. Albeit rare, such options for assessing diets, especially in studies with large sample sizes, have been used [50,51]. The food groups recognized in this study represented the foods most commonly consumed by the Russian population. They covered both healthy and unhealthy components of diets. The cross-sectional nature of the study limited the results in terms of the causal evidence for the findings. Another limitation of the study was represented by the relatively high proportion of missing data in the primary sample (11%). However, according to many authors, the strategy of removing missing data from an analysis (listwise deletion), given a sufficiently large sample size, is reasonable even if the gaps in the dataset are not random [52]. This procedure was performed in the present study.

## 5. Conclusions

Thus, the results of our study showed that individual adherence to the healthiest dietary patterns among the Russian population increased with the deterioration of social and economic living conditions, aggravation of demographic crises, higher industrial development, and increased economic inequality. Moreover, adherence to the meat-based EDP, which included a high frequency of consumption of unprocessed red meat, fish, seafood, and poultry, was higher in socially disadvantaged regions as well. The patterns revealed reflect the national preferences of Russians in their choice of foods. The effects of regional indices characterize the environmental impact on the stereotypes of eating behavior: the behavioral habits of the predominant part of the regional population are transferred onto the entire population of the region, while individual differences are leveled off to a certain extent. The results of our study represent new fundamental knowledge in terms of the impact of the living environment on human behavior, thereby correlating with previously obtained data at the population level.

## Figures and Tables

**Table 1 nutrients-15-00396-t001:** Main characteristics of the study sample.

Characteristics		Entire Sample, 18,054	Men, 6814	Women, 11,240
Place of residence, urban		79.5% (14,351)	80.1% (5459)	79.1% (8892)
Has a family		64.7% (11,680)	76.1% (5186)	57.8% (6494)
Education, higher		42.2% (7619)	43.0% (2928)	41.7% (4691)
Employment		75.7% (13,661)	83.3% (5678)	71.0% (7983)
Gastrointestinal diseases		38.2% (6902)	27.7% (1888)	44.6% (5014)
Peptic ulcer disease		12.9% (2325)	14.4% (979)	12.0% (1346)
Diabetes mellitus		4.6% (833)	3.6% (243)	5.2% (590)
Smoking	Never	61.9% (11,165)	34.0% (2316)	78.7% (8849)
	Quit	16.6% (2999)	27.7% (1885)	9.9% (1114)
	Smoker	21.5% (3890)	38.3% (2613)	11.4% (1277)
Age		46.4 ± 11.6	44.4 ± 11.9	47.5 ± 11.3
Body mass index		28.1 ± 5.9	27.6 ± 4.9	28.5 ± 6.4
Socio-geographical index		0.015 ± 0.943	0.110 ± 0.903	−0.042 ± 0.962
Demographic index		0.026 ± 0.970	0.093 ± 0.893	−0.014 ± 1.011
Industrial index		−0.015 ± 0.951	0.062 ± 0.971	−0.062 ± 0.936
Mixed index		0.045 ± 1.055	0.077 ± 1.090	0.026 ± 1.032
Economic index		−0.021 ± 0.951	−0.007 ± 0.960	−0.030 ± 0.946
Prudent dietary pattern		−0.011 ± 1.001	−0.199 ± 0.993	0.103 ± 0.989
Salt-rich dietary pattern		0.021 ± 0.991	0.223 ± 0.960	−0.102 ± 0.990
Meat-based dietary pattern		−0.004 ± 1.008	0.104 ± 0.962	−0.069 ± 1.029
Mixed dietary pattern		0.003 ± 1.007	−0.009 ± 1.008	0.011 ± 1.006
Cardioprotective dietary pattern		6.5% (1169)	4.6% (312)	7.6% (857)

**Table 2 nutrients-15-00396-t002:** Associations between individual and regional characteristics and dietary patterns (n = 18,054).

Characteristics		Prudent DP		Salt-Rich DP		Meat-Based DP		Mixed DP		Cardioprotective DP	
		B-Coeff.	*p*-Value	B-Coeff.	*p*-Value	B-Coeff.	*p*-Value	B-Coeff.	*p*-Value	OR	95% CI
Individual Characteristics											
Gender (reference: women)		−0.221	<0.001	0.219	<0.001	0.083	<0.001	0.064	0.039	0.60	0.48–0.75
Place of residence (reference: urban)		0.010	0.84	0.069	0.095	−0.056	0.34	−0.029	0.27	0.74	0.61–0.90
Has a family (reference: none)		0.057	<0.001	0.068	<0.001	0.150	<0.001	0.014	0.53	1.01	0.90–1.14
Education (reference: other than higher)		0.117	<0.001	−0.168	<0.001	0.037	0.19	0.003	0.94	1.72	1.44–2.06
Employment (reference: unemployed)		0.039	0.067	0.091	<0.001	0.104	<0.001	−0.034	0.20	1.02	0.85–1.23
Gastrointestinal diseases (reference: none)		0.088	<0.001	−0.066	<0.001	0.017	0.63	−0.060	<0.001	1.01	0.89–1.14
Peptic ulcer disease (reference: none)		−0.023	0.50	0.011	0.67	0.041	0.028	−0.027	0.19	0.81	0.65–1.02
Diabetes mellitus (reference: none)		−0.507	<0.001	−0.364	<0.001	0.207	<0.001	0.18	<0.001	1.89	1.64–2.17
Smoking (reference: never)	Quit	−0.132	<0.001	−0.004	0.81	0.030	0.17	−0.062	0.006	1.07	0.93–1.23
	Smoker	−0.317	<0.001	0.158	<0.001	0.096	<0.001	−0.129	<0.001	0.67	0.55–0.81
Age		0.002	0.11	−0.010	<0.001	−0.001	0.28	0.008	<0.001	1.02	1.01–1.03
Body mass index		−0.008	<0.001	0.005	<0.001	0.012	<0.001	0.002	<0.001	1.01	0.99–1.02
Regional Indices											
Socio-geographical index		0.095	0.076	−0.006	0.87	0.103	0.046	−0.063	0.18	1.23	1.08–1.41
Demographic index		0.039	0.37	0.043	0.22	0.039	0.42	−0.059	0.33	1.40	1.29–1.52
Industrial index		0.028	0.43	−0.025	0.31	0.059	0.12	−0.024	0.66	1.09	1.04–1.15
Mixed index		−0.054	<0.001	−0.025	0.051	0.018	0.21	0.086	<0.001	1.16	1.12–1.19
Economic index		0.009	0.86	0.013	0.67	0.077	0.20	−0.025	0.71	0.90	0.81–0.99

Adherence to the mixed EDP increased with growth in the mixed index, and the B-factor value was 0.08 at *p* < 0.001.

**Table 3 nutrients-15-00396-t003:** The values of the model effect criteria (type III likelihood-ratio test, Wald chi-squared test).

Characteristics	Prudent DP	Salt-Rich DP	Meat-Based DP	Mixed DP	Cardioprotective DP
Individual Characteristics					
Gender	41.2	94.3	10.5	4.3	19.9
Place of residence	<0.1	2.8	0.9	1.2	9.4
Has a family	13.6	17.5	50.7	0.4	<0.1
Education	33.7	117.0	1.7	<0.1	34.8
Employment	3.3	15.8	61.3	1.6	<0.1
Gastrointestinal diseases	16.1	10.8	0.2	7.3	<0.1
Peptic ulcer disease	0.4	0.2	4.8	1.7	3.3
Diabetes mellitus	97.8	45.0	31.6	16.7	79.8
Smoking	181.0	38.7	24.3	25.5	23.7
Age	2.5	53.8	1.1	59.0	15.9
Body mass index	110.7	13.2	51.4	0.5	0.7
Regional Indices					
Socio-geographical index	3.2	<0.1	4.0	1.8	9.7
Demographic index	0.8	1.5	0.6	1.0	64.2
Industrial index	0.6	1.0	2.4	0.2	11.0
Mixed index	26.6	3.8	1.5	19.3	84.2
Economic index	<0.1	0.2	1.6	0.1	4.6

## Data Availability

The datasets presented in this article are not readily available because of the prohibition of data transfer to third parties. Requests to access the datasets should be directed to Svetlana Shalnova at svetlanashalnova@yandex.ru.

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
