# Peer review of "Regional Living Conditions and Individual Dietary Characteristics of the Russian Population"

_nutrients, 2023, doi:10.3390/nu15020396_

Round 1

Reviewer 1 Report

This study aimed to evaluate the effect of regional characteristics of the living environment on a priori and a posteriori dietary patterns of individuals from the Russian population. 

In general, the research question is interesting, the sample population is fairly representative, the methodology is generally well-described and conducted, and the results are clearly described. However, there is a major concern related to one of the authors' main findings:

-       The statements and conclusions (lines 229-237; 269-271; 298-311; 370-373) about the probability that a cardioprotective healthy dietary pattern (CPDP) increases with deteriorating social living conditions (socio-geographical index) because frequent consumption of healthy foods is available and preferable for all segments of the population seem rather generalistic and somewhat contradictory to previous author's conclusions (Karamnova NS et al Russ J Cardiol 2020). In fact, among the regional indices considered herein (which provide information on living conditions that turned out to be more representative among 12 regions of the Russian Federation), the socio-geographical index accounted for the largest percentage (28%) of the total variance (Maksimov S et al, Int J Public Health 2021). While, on the other hand, the CPDP was not highly prevalent among the Russian population studied (6.5% and 7%, in present and previous study), being more frequent among women, urban residents, people with higher education and unemployed participants. It makes little sense to have included in this study a dietary pattern that was previously observed to be infrequent among the population. The associations observed herein could have resulted by chance.

Minor comments:

-       Introduction: Authors contextualize the importance of their study by arguing that identifying regional environmental predictors (characteristics of the living environment) that may contribute to different dietary patterns among the Russian population may represent an opportunity to understand and, hopefully, modify the underlying causes of the unequal distribution of population health. It is suggested to highlight background information on the main diet-related health concerns of the Russian population.  

Is there a known healthy traditional Russian dietary pattern or diet?

-       Lines 130-131: Give details on why particularly these pathologies. Specify, diabetes mellitus type I, type II or both.

-       Line 206: “the share of those a family” is not clear

Author Response

We thank the reviewer for commenting on our article.

Let us address your comments:

  1. The statements and conclusions (lines 229-237; 269-271; 298-311; 370-373) about the probability that a cardioprotective healthy dietary pattern (CPDP) increases with deteriorating social living conditions (socio-geographical index) because frequent consumption of healthy foods is available and preferable for all segments of the population seem rather generalistic and somewhat contradictory to previous author's conclusions (Karamnova NS et al Russ J Cardiol 2020). In fact, among the regional indices considered herein (which provide information on living conditions that turned out to be more representative among 12 regions of the Russian Federation), the socio-geographical index accounted for the largest percentage (28%) of the total variance (Maksimov S et al, Int J Public Health 2021). While, on the other hand, the CPDP was not highly prevalent among the Russian population studied (6.5% and 7%, in present and previous study), being more frequent among women, urban residents, people with higher education and unemployed participants. It makes little sense to have included in this study a dietary pattern that was previously observed to be infrequent among the population. The associations observed herein could have resulted by chance.

Our response:

The conclusion about the high frequency of consumption of healthy foods (presumed by CPDP) by the Russian population is not entirely consistent with the relatively low frequency of CPDP per se in the population – only at first glance. The fact is that CPDP is the most ‘stringent’ option of a healthy diet, including all four components, and due to its stringency, it is not often detected among the population. The associations of CPDP with regional indices, obtained in our study, only reflect the following trend in the dietary preference of the population: with deterioration of social living conditions, the likelihood of consuming healthy foods (encompassed by CPDP) increases.

In order to demonstrate that the obtained associations are consistent and accurately reflect population dietary patterns, we conducted an additional analysis of the association of regional indices with a three-component CPDP. That is, we considered an alternative dietary pattern when an individual adheres to at least three of the four following eating habits: daily consumption of fruit and vegetables, eating fish at least 1-2 times a week, the exclusive use of vegetable oils for cooking, and consumption of dairy products with reduced fat content. Consequently, the ‘stringency’ of CPDP decreased, but its incidence in the population increased to 32.6%. The associations of the three-component CPDP with the socio-geographical, demographic and mixed indices remained direct and statistically significant, which, in our opinion, is indicative of the validity of our initial conclusion. To clarify these issues, we have supplemented our article with an additional table, Table S3 (Associations of individual and regional characteristics with a three-component cardioprotective dietary pattern). Also, we have added essential explanations to the Materials and Methods (lines 131-137) and Results sections (lines 255-260).

  1. Introduction: Authors contextualize the importance of their study by arguing that identifying regional environmental predictors (characteristics of the living environment) that may contribute to different dietary patterns among the Russian population may represent an opportunity to understand and, hopefully, modify the underlying causes of the unequal distribution of population health. It is suggested to highlight background information on the main diet-related health concerns of the Russian population.  

Is there a known healthy traditional Russian dietary pattern or diet?

Our response:

We have added information about the main dietary patterns of the Russian population and their associations with health indicators (risk factors for chronic non-communicable diseases) (lines 68-77).

  1. Lines 130-131: Give details on why particularly these pathologies. Specify, diabetes mellitus type I, type II or both.

Our response:

We have corrected the text (lines 145-148).

  1. Line 206: “the share of those a family” is not clear

Our response:          

We have corrected the text (lines 223-224).

Reviewer 2 Report

    The study by Sergey A. Maksimov el. al demonstrated the effect of regional characteristics of living environment on individual and dietary patterns of the Russian population. Here, they found adherence to the meat-based EDP was directly associated with deterioration of social living conditions and the more northerly location of the region of residence. The revealed patterns reflect the national dietary preferences of Russians, and regional indices characterize the effect of living environment. 

      Overall, this is an interesting study which revealed the national dietary preferences of Russians. However, the English writing of this study needs to be improved for publication.

Author Response

We thank the reviewer for commenting on our article.

Let us address your comments:

Overall, this is an interesting study which revealed the national dietary preferences of Russians. However, the English writing of this study needs to be improved for publication.

Our response:

In compliance with your suggestion, the text of the manuscript was carefully proofread by a professional translator formerly associated with Yale University and North Carolina State University. Accordingly, a number of corrections have been made to the entire text of the article.

Reviewer 3 Report

Cancer is  one of the leading causes of death in Russia, second only to deaths from cardiovascular diseases. In 2019, more than 158,000 men and 136,000 women died of cancer. The latest available data show that, in 2020, the most common cancer by incidence in Russia was colorectal cancer, followed by breast cancer and lung cancer. Incidence rates of colorectal cancer have consistently increased over the past 3 decades (from 16.9 to 24.3 per 100,000 women and 22.7 to 35.3 per 100,000 men between 1993 and 2019), reflecting changes in the prevalence of risk factors, diet an aging population,

Although stomach cancer was the second most common cause of cancer-related deaths in the later part of the 20th century, it has now been replaced by colorectal cancer.  Predicted trends show that the number of cancer-related deaths is expected to rise 19.5% over the next 2 decades, from around 312,000 deaths in 2020 to 373,000 deaths in 2040 . The paper is interesting but I believe to add data on cancer rate to better understand the relationship between diet and the increase of cancer rate

Author Response

We thank the reviewer for commenting on our article.

Let us address your comments:

Cancer is one of the leading causes of death in Russia, second only to deaths from cardiovascular diseases. In 2019, more than 158,000 men and 136,000 women died of cancer. The latest available data show that, in 2020, the most common cancer by incidence in Russia was colorectal cancer, followed by breast cancer and lung cancer. Incidence rates of colorectal cancer have consistently increased over the past 3 decades (from 16.9 to 24.3 per 100,000 women and 22.7 to 35.3 per 100,000 men between 1993 and 2019), reflecting changes in the prevalence of risk factors, diet an aging population,

Although stomach cancer was the second most common cause of cancer-related deaths in the later part of the 20th century, it has now been replaced by colorectal cancer.  Predicted trends show that the number of cancer-related deaths is expected to rise 19.5% over the next 2 decades, from around 312,000 deaths in 2020 to 373,000 deaths in 2040. The paper is interesting but I believe to add data on cancer rate to better understand the relationship between diet and the increase of cancer rate

Our response:

Indisputably, the relationship between the incidence of cancer and the dietary preferences of the population is of high scientific interest. However, the goal of our study was different: to reveal the dependence of individual a priori and a posteriori dietary patterns of the Russian population on the regional characteristics of living environment. We believe that adding health data to the article (including prevalence of cancer) will not add anything useful to the manuscript in the light of the stated goal of the study.

Round 2

Reviewer 1 Report

The additional analyses performed by the authors confirm their previous associations and findings. 

Comments and suggestions were satisfactorily corrected.

Reviewer 3 Report

Comments that you do not cover "for outside the scope of the paper" could have further improved the overall quality of the work, but I accept your decision.